# Identification of Immune Markers in Dilated Cardiomyopathies with Heart Failure by Integrated Weighted Gene Coexpression Network Analysis

**DOI:** 10.3390/genes13030393

**Published:** 2022-02-22

**Authors:** Xuehua Wang, Hongquan Guan, Wei Liu, Huili Li, Jiaxing Ding, Yu Feng, Zhijian Chen

**Affiliations:** Department of Cardiology, Wuhan Union Hospital, Tongji Medical College, Huazhong University of Science and Technology, No. 1277 Jiefang Avenue, Wuhan 430022, China; wxh2020@hust.edu.cn (X.W.); guan-hong-quan@163.com (H.G.); liuwei2019@hust.edu.cn (W.L.); lihuili09251216@163.com (H.L.); jxding@hust.edu.cn (J.D.); 15036067026@163.com (Y.F.)

**Keywords:** dilated cardiomyopathy, heart failure, immunity, GEO, WGCNA, CIBERSORT

## Abstract

Dilated cardiomyopathy (DCM), a heterogeneous cardiomyopathy, is a major cause of heart failure and heart transplant. Currently, immunotherapy is believed to be an effective treatment method for DCM. However, individual differences are so obvious that the clinical effect is not satisfactory. In order to find immune-related biomarkers of DCM to guide treatment and improve clinical efficacy, we downloaded a GSE120895 dataset from the Gene Expression Omnibus (GEO) database using CIBERSORT and WGCNA algorithms in RStudio and visualizing the protein–protein interaction (PPI) network for key modules by Cytoscape, and finally obtained six hub genes. A GSE17800 dataset was downloaded from the GEO dataset to verify the diagnostic values of hub genes, *MYG1*, *FLOT1*, and *ATG13*, which were excellent. Our study revealed unpublished potential immune mechanisms, biomarkers, and therapeutic targets of DCM.

## 1. Introduction

Heart failure (HF) is a clinical syndrome with typical symptoms, such as breathlessness and ankle swelling, resulting from cardiovascular or noncardiovascular disease [1]. There are approximately 64.3 million people worldwide suffering from HF, which plays a vital role in the global burden of diseases, and a trend of younger HF patients needs our attention [2]. The authors of [3] found that although the age-specific mortality was decreased, the absolute number of deaths increased owing to population growth and aging based on mortality data from the Global Burden of Disease Study 2013.

Dilated cardiomyopathies (DCMs), one of the leading causes of HF, are a heterogenous heart muscle disease without ischemia characterized by systolic dysfunction and ventricular dilatation, which has a rising prevalence reaching 1/250 but with a poor prognosis [4]. Failure of the heart is the most important pathway of DCM identified by ingenuity pathway analysis (IPA) in a meta-analysis [5]. The pathophysiology of HF in DCM is complicated, and immune activation in the myocardium after virus infection is considered to be the main reason for this [6]. Barth [7] studied the identification of a common gene expression signature in DCM across two different studies, showing that the most obvious downregulation was the immune response process in both microarray types. Additionally, a number of autoantibodies against different cardiomyocyte proteins in DCM have been found [8]. Owing to the disorder of the immune system in DCM, immunotherapy relying on precision medicine came into being, including immunosuppressants and immunoadsorption [9]. However, an increase in antibodies and a decrease in contractility were observed in DCM 12 months after immunoadsorption therapy and subsequent IgG replacement (IA/IgG) treatment [10]. The effectiveness of immunotherapy needs to be further evaluated. Moreover, it is worth exploring which kind of patients can reap the maximum benefit from immunotherapy.

In recent years, the vigorous development of high-throughput biological analysis technology has promoted the explosive growth of genomic biological data, and the application of informatics in biomedical research has been increasing [11]. The systematic method in biology breaks the traditional analysis of a single gene or multiple genes at a time, giving rise to a totally new level of gene research, with which we can identify the gene expression profiles of complex diseases [12]. Gene expression provides a key link in the prognosis of disease. Laura [13] proposed, for the first time, that it was possible to judge the efficacy of adjuvant systemic therapy in breast cancer patients by analyzing DNA microarrays, patients who would benefit from which could be selected for treatment, greatly improving the efficiency and reducing the cost of treatment. CIBERSORT is an analytical tool of immune infiltration developed by the Stanford University research team and published in the *Nature Methods* in 2015 using a deconvolution algorithm to estimate the composition and abundance of the immune cells in the mixed cells based on transcriptome data [14]. Weighted gene coexpression network analysis (WGCNA) is another biological method performing relevant network analysis on the dataset to obtain biomarkers or therapeutic targets [15]. Based on the above methods, we aimed to use gene expression data from the Gene Expression Omnibus (GEO) dataset to reveal immune-related biomarkers of DCM and provide assistance in assessing the effectiveness of immunotherapy in individuals (Figure 1).

## 2. Materials and Methods

### 2.1. Data Processing and Analysis

We downloaded the GSE120895 [16] and GSE17800 [17] datasets from GEO (URL: https://www.ncbi.nlm.nih.gov/geo/ (accessed on 17 January 2022)) using RStudio 4.1.1 software (URL: https://www.r-project.org/ (accessed on 2 September 2021)). Subsequent data processing and analysis were all performed in RStudio. The GSE120895 dataset (GPL570 platform), which was updated in 2021, was used for transcriptome analyses of the endocardium myocardia of 47 DCM patients and 8 individuals with normal left ventricular ejection fraction (LVEF). The GSE17800 dataset (GPL570 platform) found 40 endocardial samples from DCM and 8 normal control endocardial samples. Both GSE120895 and GSE17800 comprised DCM patients with left ventricular systolic dysfunction (LVEF < 45%) and symptoms of HF according to New York Heart Association (NYHA) classifications Ⅱ and Ⅲ. Gene expression values were log2 transformed and normalized by “limma” package [18]. We used the “hgu133plus2” package to convert to gene symbols and finally obtained a gene expression profile containing 12,412 genes from 55 samples of GSE120895. Differentially expressed genes (DEGs) were filtered by |Log2 fold change| > mean plus twice the standard deviation and *p*-value < 0.05.

### 2.2. Functional Enrichment Analysis

Kyoto Encyclopedia of Genes and Genomes (KEGG) pathway enrichment analysis for all expression data was performed by gene set enrichment analysis (GSEA) with the “clusterProfiler [19]” and “DOSE” packages [20] in RStudio, setting the *p*-value to less than 0.05 and the number of permutations to 1000.

### 2.3. CIBERSORT

In order to determine the differences in the composition of immune cells between DCM with HF and normal individual controls, CIBERSORT was used to analyze DEGs. We used the LM22 signature and 1000 permutations in RStudio to obtain the proportion of each type of immune cell in the samples of two groups [21]. We visualized the results obtained above using the “ggplot2”, “ggpubr”, and “heatmap” packages in RStudio.

### 2.4. WGCNA

WGCNA in RStudio was introduced to construct a coexpression network (undirected weighted gene networks [15]) with the top 5000 genes with the highest median absolute deviation [22]. We constructed the adjacency matrix with a suitable soft threshold (ß = 17) when R^2^ reached 0.85 after calculating the Pearson correlation coefficient between any two genes in the immune cells. We combined similar modules (modules are clusters of highly interconnected genes [15]) with a similarity of 0.85 and then identified modules related to immune cells.

### 2.5. Protein–Protein Interaction (PPI) Network Analysis and Obtained Hub Genes

Cytoscape (v3.8.2, URL: https://cytoscape.org/ (accessed on 4 September 2021)) was used to visualize the PPI network (a tool to describe a biological system according to proteins and the relationships [23]) of the top 300 genes with a weight value of the key modules. Hub genes (highly connected genes [15]) were selected through the Cytoscape plugin cytoHubba [24], and we chose 3 hub genes from each module according to the largest degree.

### 2.6. Validation of Hub Genes

For the identification of hub genes, we used the “pROC” package to draw receiver operating characteristic (ROC) curves and calculated the area under the ROC curve (AUC) in GSE17800 [25]. AUC > 0.7 indicated that the gene had a good fitting effect, and *p* < 0.05 indicated statistical significance. We further confirmed the expression values of hub genes in each group.

## 3. Results

### 3.1. Identification of DEGs

GSE120895 from the GEO dataset contained 12,412 genes in 55 samples of endocardium myocardium from 47 DCM patients with HFrEF and 8 individuals with normal LVEF. A total of 473 genes were identified as DEGs in GSE120895, among which 268 genes were upregulated and 205 genes were downregulated, which is shown in a volcano plot (Figure 2A). A heatmap was used to visualize the top 25 genes of DEGs (Figure 2B).

### 3.2. Functional Correlation Analysis

The results of KEGG from GSEA (Figure 3B) showed that the KEGG pathways were enriched in human papillomavirus infection, focal adhesion, the PI3K-Akt signaling pathway, the cGMP-PKG signaling pathway, viral carcinogenesis, Epstein–Barr virus infection, and herpes simplex virus 1 infection, participating in the process of viral infection and immunity. Most of them were at the top (Figure 3A).

### 3.3. Immune Cell Infiltration

There was a significant difference between the DCM with HF and controls in the composition of immune cells (Figure 4). CD4 memory resting T cells (*p* = 0.035), CD8 T cells (*p* = 0.05), CD4 naive T cells (*p* = 0.015), plasma cells (*p* = 0.0047), macrophages M0 (*p* = 0.059), and NK resting cells (*p* = 0.017) were significantly higher in the DCM with HF. Otherwise, B cells naive (*p* = 0.015), B cells memory (*p* = 0.0074), T cells follicular helper (*p* = 7.1 × 10^−7^), and CD4 memory activated T cells (*p* = 1.3 × 10^−6^) were much lower in the DCM patients with HF.

### 3.4. Construction of Weighted Coexpression Network and Identification of Immune-Corrected Modules

In this study, after filtering an outlier (Figure 5A), ß = 17 (scale-free R^2^, 0.85) was chosen as the soft threshold to construct a scale-free network, while the network topology with 1–30 threshold weights was analyzed (Figure 5B). We merged the modules with similarity greater than 0.85 and finally obtained six modules (Figure 5C). The correlation between the modules and 22 kinds of immune cells was shown in a network heatmap (Figure 5D), which indicated that the blue and black modules were the key modules. Correlation analysis was performed between module eigengenes (MEs, the first principal component of a module [15]) and immune cells. Blue and black modules significantly correlated with NK resting cells and B cells (Figure 5E), which were further validated by the scatter plot (Figure 6).

### 3.5. PPI Network Construction and Identification of Hub Genes

The top 300 genes with weight values of the key modules were introduced into Cytoscape to establish PPI networks, and 10 hub genes are shown in different colors (Figure 7). We selected six hub genes according to the highest degree: *MYG1* (degree = 104) and *FLOT1* (degree = 104) with *GPX1* (degree = 47) in the black module, and *LINC00520* (degree = 59), *ZNF548* (degree = 52), and *ATG13* (degree = 36) in the blue module.

### 3.6. ROC Curve Analysis of Hub Genes

We drew the ROC curve of hub genes in RStudio so as to clarify the diagnostic value. The results showed *MYG1* (AUC = 0.741), *FLOT1* (AUC = 0.766), *GPX1* (AUC = 0.672), *LINC00520* (AUC = 0.716), *ZNF548* (AUC = 0.706), and *ATG13* (AUC = 0.806) in the GSE17800 validation set (Figure 8), out of which *MYG1*, *FLOT1*, and *ATG13* (*p* < 0.05) had significant diagnostic values. The gene expression level provided further proof (Figure 9), showing that *MYG1*, *FLOT1*, and *ATG13* were all upregulated.

## 4. Discussion

The Prospective Urban Rural Epidemiology (PURE) cohort study [26] published in *Lancet* suggests that, among adults aged 35–70 around the world, cardiovascular disease is still the major cause of death, and heart failure is one of main events leading to mortality. While the global burden of HF is quite heavy with a stubbornly high prevalence and mortality rate, targeted therapy (precision medicine) based on high-throughput molecular biology technology may provide a new direction [27].

Immune response as a defense mechanism results in long-term myocardial immune infiltration, which can cause ventricular remodeling in HF patients [28]. Evidence suggests that HF is related to the infiltration of immune cells in the myocardium [29]. DCM is one of the most common causes of HF worldwide, with myocardial injury mediated by multiple factors, such as familial susceptibility, infection, immunity, toxicants, endocrine, and metabolic abnormalities [30]. Abnormalities of immune cells are considered to connect with the outcome of DCM [31,32]. In addition, Staudt [33] found that the disturbance of humoral immunity might promote the development of HF, and it was identified that there were a number of autoantibodies against cardiac cell proteins in DCM, some of which were related to HF [34]. In a word, humoral and cellular immune disorders in DCM are gradually uncovered, playing an important role in the pathogenesis. With bioinformatics algorithms and tools, we can explore the immune mechanism of DCM with HF and find immune markers of DCM.

In our study, the results of KEGG from GSEA suggested that the KEGG pathways were enriched in the process of viral infection and immunity. We attempted to determine the immune-related pathological process and marker genes in DCM with WGCNA and CIBERSORT algorithms based on the GSE120895 dataset from the GEO database. The composition of immune cells in two groups was totally different. The high expression of NK cells and CD8 T cells in DCM with HF indicated that the cellular immunity was active. CD4 memory-activated T cells, naive B cells, memory B cells, and follicular helper T cells were clearly reduced in DCM, while plasma cells significantly increased, which meant that the humoral immune system was abnormal in DCM. CD4 T cells are closely involved in the progression of cardiac insufficiency [35]. The authors of [36] found that the number of NK cells in DCM was significantly more than that in the control in the 1990s, and functional abnormalities of NK cells might contribute to pathogenesis in DCM. Recently, immunoadsorption has become a new treatment option for DCM, but the clinical effect has not met expectations, and individual differences are obvious due to uncertain reasons [37]. Bhardwaj [38] attempted to distinguish between DCM patients who responded and those who did not respond to immunoadsorption at the proteomics level, and they found that the proteins S100-A8, perilipin-4, and kininogen-1 had the potential to help stratify patients with immunoadsorption therapy. We aimed to identify DCM patients who may respond to immunotherapy at the genetic level so as to improve the effectiveness of immunotherapy and achieve precision medicine.

We constructed six modules correlated with immune by WGCNA analysis, in which blue and black modules showed a strong correlation with NK resting cells, B cells, and plasma cells. We finally obtained six hub genes, *MYG1*, *FLOT1*, *GPX1*, *LINC00520*, *ZNF548*, and *ATG13*, from blue and black modules. *MYG1*, *FLOT1*, and *ATG13* were verified in GSE17800 (AUC > 0.7, *p* < 0.05).

*MYG1 (melanocyte proliferating gene 1)* located in the nucleus and mitochondria has exonuclease activity, mainly controlling mitochondria’s functions [39]. As a ubiquitously expressed and highly conserved factor, *MYG1* is involved in immune regulation [40], which is not only associated with leukoplakia susceptibility, but also promoting the progression of lung adenocarcinoma and inhibiting autophagy [41]. Compared with the control, *MYG1* was differentially expressed in DCM in our study. We suppose that the involvement of *MYG1* in DCM progressed to HF, which requires further research to confirm.

*FLOT1 (flotillin-1)*, another ubiquitously expressed and highly conserved raft-associated protein, maintains the membrane integrity of B and T lymphocytes [42], playing a role in T-cell activation [43]. *Flotillin-1* was reported to be involved in cell adhesion, the overexpression of which enhanced cell spreading [44]. *FLOT1*, with a great diagnosis value shown by the ROC curve, was overexpressed in DCM with HF, and we also found that focal adhesion was one of the significant enrichment pathways in our study. Based on the above, we could make a hypothesis that *FLOT1* has an effect on the development of DCM by activating T cells and speeding up cell spread.

As a key component of *ULK1 (unc-51 like autophagy activating kinase 1)*, *ATG13 (autophagy-related protein 13)* is responsible for activating *ULK1* kinase, which is closely associated with autophagy, which may have a connection with inflammation [45]. Autophagy proteins participate in the development and homeostasis of the immune system, antigen presentation, and regulation of immune signals, playing an essential role in controlling inflammation [46]. Autophagy may take part in the development of HF in DCM [47]. ROC curve analysis showed that the AUC of *ATG13* was greater than 80%, which, with increased expression, could provide a sign of immune disorders in DCM.

Ischemic cardiomyopathy (ICM), which is another disease causing inflammation in the heart, has some difference in the enriched pathway from DCM. The cytoskeletal and immune pathways belong to ICM, while the adhesion pathway is enriched in DCM [48]. Aiqing [49] found that upregulated DEGs in DCM or ICM were not associated with each other, and the hub genes in our study were not on the list of common regulated genes from Aiqing’s study. ICM was excluded by angiography in GSE120895 and GSE17800 datasets, so the hub genes in ICM may not be upregulated, while arrhythmogenic right ventricular cardiomyopathy (ARVC), characterized by the fibrofatty replacement of myocardium, has more active inflammatory signaling compared with DCM [50]. ARVC and DCM differ markedly at the transcriptomic level [51]. However, it cannot be ruled out that *MYG1*, *FLOT1*, and *ATG13* are upregulated in ARVC. Further experiments can identify genetic changes in ICM and ARVC.

Immune-related *MYG1*, *FLOT1*, and *ATG13* had excellent diagnostic values. We may determine the success rate of immunotherapy by measuring the expression of these genes and increasing clinical effectiveness.

## 5. Conclusions

In summary, we used the WGCNA and CIBERSORT bioinformatics algorithms to identify immune biomarkers in DCM with HF, and obtained two key modules and six hub genes, out of which *MYG1*, *FLOT1*, and *ATG13*, with good diagnostic values, may be potential diagnostic biomarkers and therapeutic targets. We speculate that these genes have the ability to predict the success rate of immunotherapy, which needs to be confirmed by further experiments.

## Figures and Tables

**Figure 1 genes-13-00393-f001:**
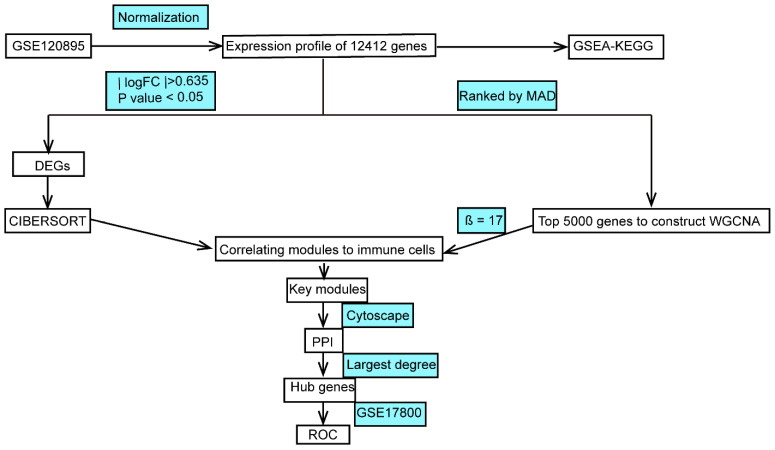
Study flowchart (GSEA, gene set enrichment analysis; KEGG, Kyoto Encyclopedia of Genes and Genomes; MAD, median absolute deviation; DEGs, differentially expressed genes; WGCNA, weighted gene coexpression network analysis; PPI, protein–protein interaction; ROC, receiver operating characteristic).

**Figure 2 genes-13-00393-f002:**
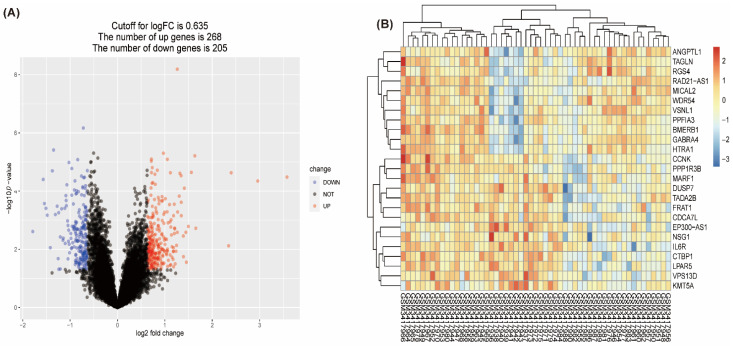
Volcano plot and heatmap for differentially expressed genes identified in the GSE120895 dataset; red indicates upregulated genes, and blue indicates downregulated genes. (**A**) The volcano plot shows up- and downregulated genes. (**B**) The heatmap shows the expression of the 25 most differentiated genes in individuals.

**Figure 3 genes-13-00393-f003:**
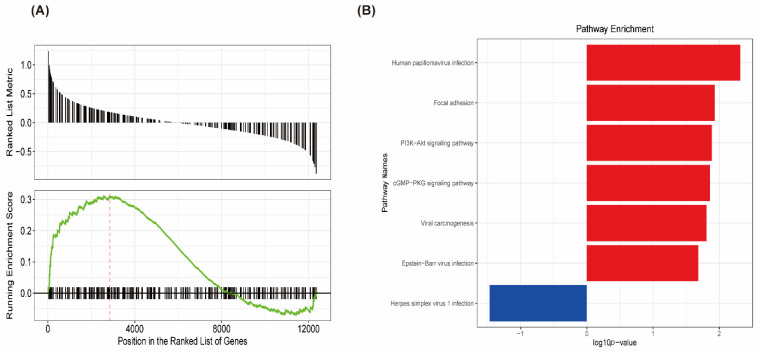
Gene set enrichment analysis (GSEA) and Kyoto Encyclopedia of Genes and Genomes (KEGG) pathway enrichment analysis for GSE120895. (**A**) GSEA revealed the genes were enriched at the top. (**B**) The top terms of the KEGG pathway analysis for GSE120895. The significant pathways are represented by the negative decimal logarithm of the *p*-value.

**Figure 4 genes-13-00393-f004:**
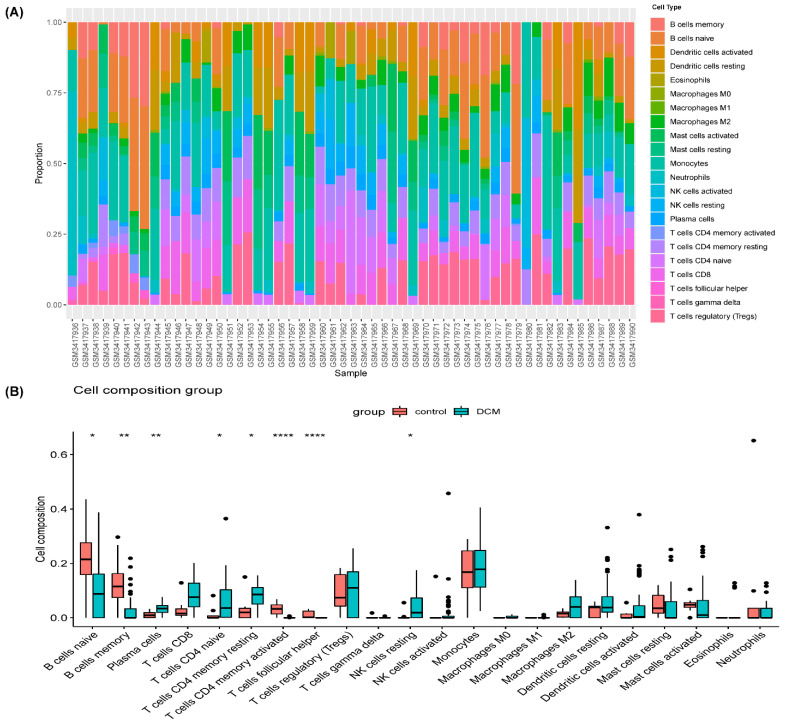
Immune infiltration analysis for GSE120895. (**A**) Bar charts of 22 immune cell proportions in DCM and control tissues. (**B**) Differential expression of immune cells in two groups. * *p* < 0.05, ** *p* < 0.01, **** *p* < 0.0001.

**Figure 5 genes-13-00393-f005:**
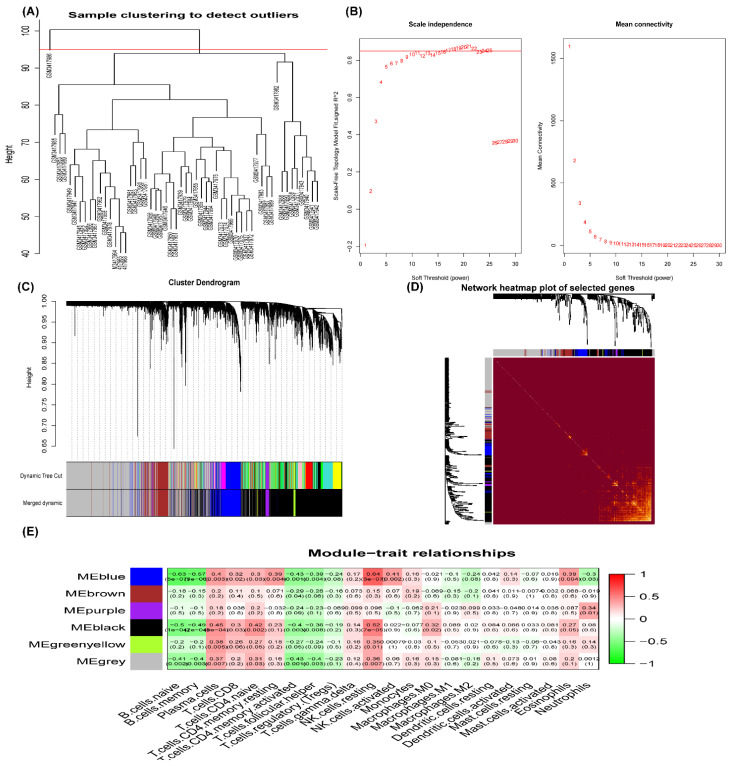
Weighted gene coexpression network analysis (WGCNA) for GSE120895. (**A**) The sample clustering to detect an outlier. (**B**) Analysis of the scale-free index (left) and mean connectivity (right) for various threshold powers. ß = 17 (scale-free R^2^, 0.85) was chosen as the soft threshold. (**C**) Clustering dendrogram of genes, together with assigned module colors. Merge modules with a similarity of 0.85. (**D**) Network heatmap plot in the coexpression modules. Light color represents higher overlap. (**E**) Module–trait relationships; the numbers in the cell represent the correlation coefficient and corresponding *p*-value. The correlation coefficient is between −1 and +1, and the larger the absolute value, the stronger the association.

**Figure 6 genes-13-00393-f006:**
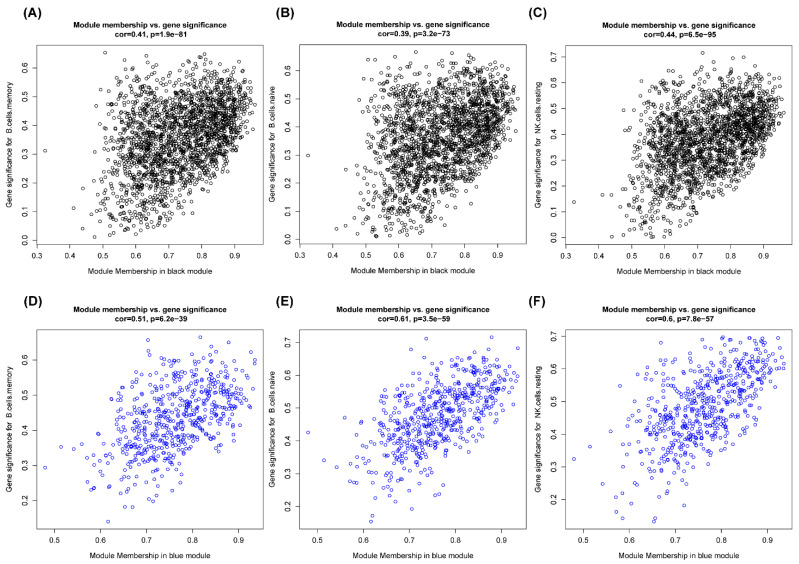
The scatter plots of module membership and gene significance for B cells and NK cells in the key modules. One dot represents one gene. (**A**) ROC curves of *MYG1*. (**B**) ROC curves of *FLOT1*. (**C**) ROC curves of *GPX1*. (**D**) ROC curves of *LINC00520*. (**E**) ROC curves of *ZNF548*. (**F**) ROC curves of *ATG13*. AUC > 0.7 indicates good effect.

**Figure 7 genes-13-00393-f007:**
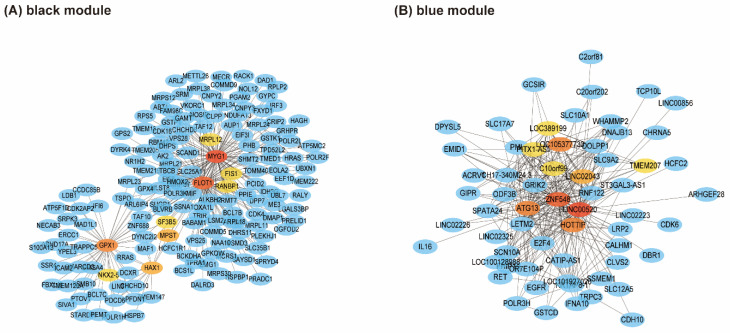
Key modules that contain hub genes of the PPI network. Hub genes are shown in different colors (the darker the gene, the higher the score). (**A**) PPI network for black module. (**B**) PPI network for blue module. *MYG1* (degree = 104), *FLOT1* (degree = 104), *GPX1* (degree = 47), *LINC00520* (degree = 59), *ZNF548* (degree = 52), and *ATG13* (degree = 36) were selected as hub genes.

**Figure 8 genes-13-00393-f008:**
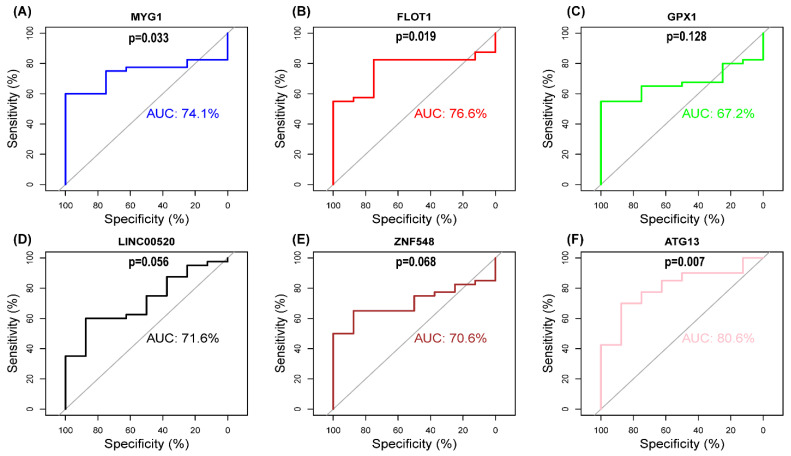
Receiver operating characteristic (ROC) curves of hub genes in GSE17800. (**A**) ROC curves of *MYG1*. (**B**) ROC curves of *FLOT1*. (**C**) ROC curves of *GPX1*. (**D**) ROC curves of *LINC00520*. (**E**) ROC curves of *ZNF548*. (**F**) ROC curves of *ATG13*. AUC > 0.7 indicates good effect.

**Figure 9 genes-13-00393-f009:**
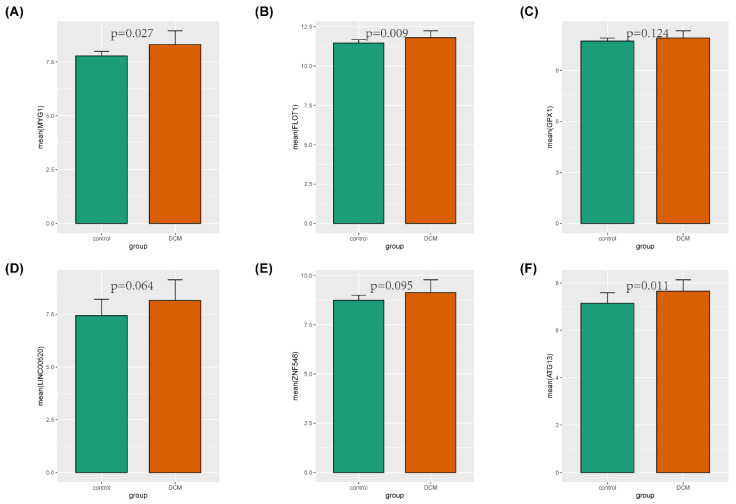
GSE17800 was used to verify the expression levels of hub genes in DCM and control. (**A**) Gene expression values of *MYG1* among samples. (**B**) Gene expression values of *FLOT1* among samples. (**C**) Gene expression values of *GPX1* among samples. (**D**) Gene expression values of *LINC00520* among samples. (**E**) Gene expression values of *ZNF548* among samples. (**F**) Gene expression values of *ATG13* among samples.

## Data Availability

The data presented in this study are openly available in Gene Expression Omnibus (GEO) (URL: https://www.ncbi.nlm.nih.gov/geo/, accessed on 17 January 2022).

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
