# Peer review of "Identification of Immune Markers in Dilated Cardiomyopathies with Heart Failure by Integrated Weighted Gene Coexpression Network Analysis"

_genes, 2022, doi:10.3390/genes13030393_

Round 1
Reviewer 1 Report
This manuscript analyzed microarray expression data to identify immune-related biomarkers of dilated cardiomyopathy. While the overall study design is convincing and the outcome seem valuable, the manuscript’s text and figures require significant revision. Some suggestions are given below.
- The quality of English needs improvement. Here are some examples: “…, one of the leading causes of HF, is a heterogenous of heart muscle disease …”, “Owning to the disorder of the immune system in DCM” [should be owing], “precision medicine came to being, including immunosuppressants and immunoadsorption” [the sentence is unclear]
- Several figures are not legible. For example, Figs 2, 3B, 5D, 7 are not publication quality.
- Need a better discussion of why the authors considered this dataset as the most informative dataset for their purpose. Similarly, the choice of validation dataset needs a better rationale.
- The authors assumed readers will have a working knowledge of WGCNA, GSEA, and cytoscape, which is not a fair assumption and hurts the manuscript’s readability. Concepts like eigengene need better explanation.
- The manuscript did a large-scale data analysis, but did not follow many of the “systems biology” norms. Cut-offs were chosen arbitrarily without showing the underlying distribution. For example, what is the degree distribution of nodes shown in Fig 7A?
Author Response
Point 1. The quality of English needs improvement. Here are some examples: “…, one of the leading causes of HF, is a heterogenous of heart muscle disease …”, “Owning to the disorder of the immune system in DCM” [should be owing], “precision medicine came to being, including immunosuppressants and immunoadsorption” [the sentence is unclear]
Response 1: Thanks for your suggestion. The manuscript has underwent English revisions to improve the English quality.
Point 2. Several figures are not legible. For example, Figs 2, 3B, 5D, 7 are not publication quality.
Response 2: Thanks for your suggestion. We apologize for providing substandard figures. We had made the figures smaller to present the result in one figure, which results in poor font resolution. We made the figures larger (Figs 2, 3B and 5D, Line 129, 140 and 167). We made a new figure 7 instead of the old one to present the result well (Line 180).
Point 3: Need a better discussion of why the authors considered this dataset as the most informative dataset for their purpose. Similarly, the choice of validation dataset needs a better rationale.
Response 3: Thank you for the excellent suggestion. We have made revisions according to your comments (Line 73-79). The following reasons we choose GSE120895 for study: (a) GSE120895 was used for transcriptome analyses of the endocardium myocardia of 47 DCM patients and 8 individuals with normal left ventricular ejection fraction (LVEF), which had enough samples; (b) The dataset was public on Oct 05, 2021 in GEO. And the GSE17800 also contained 48 samples, which included 8 normal control endocardial samples. Both GSE120895 and GSE17800 comprised DCM patients with left ventricular systolic dysfunction (LVEF<45%) and symptoms of HF according to the New York Heart Association (NYHA) classifications â…¡ and â…¢. Therefore, We chose them for our study.
Point 4. The authors assumed readers will have a working knowledge of WGCNA, GSEA, and cytoscape, which is not a fair assumption and hurts the manuscript’s readability. Concepts like eigengene need better explanation.
Response 4: We gratefully appreciate for your valuable comment. We have revised the manuscript according to your comments, giving explanation for co-expression network (Line 98), modules (Line 101), PPI network (Line 106), hub genes (Line 107) and eigengenes (Line 163).
Point 5: The manuscript did a large-scale data analysis, but did not follow many of the “systems biology” norms. Cut-offs were chosen arbitrarily without showing the underlying distribution. For example, what is the degree distribution of nodes shown in Fig 7A?
Response 5: Thanks for your valuable suggestion. We have read a lot of papers before the data analysis. At the first time, we planned to use DEGs of GSE120895 for WGCNA and CIBERSORT analysis and the main results are shown in Figure â‘ , which showed blue and brown modules significantly correlated with NK resting cells and B cells. However, we were concerned that the results were less reliable because there were only 473 genes in DEGs. Therefore, WGCNA in R Studio was introduced to construct a co-expression network with the top 5000 genes with the highest median absolute deviation (Lu L et al., 2021). Blue and black modules were identified as key modules which were used for PPI network analysis, and we finally got 6 hub genes according to the highest degree: MYG1 (degree=104) and FLOT1 (degree=104) with GPX1 (degree=47) in the black module, and LINC00520 (degree=59), ZNF548 (degree=52), and ATG13 (degree=36) in the blue module (Line 187-189). HAX1 (degree=15) in black module had the fourth highest degree, which is much smaller than MYG1 (degree=104) and FLOT1 (degree=104) with GPX1 (degree=47) , so we only selected 3 hub genes from each module to verify. We added the degree of nodes in Fig 7 legends (Line 183-184).
(please see figure ①in attachment)

Reviewer 2 Report
Dilated cardiomyopathy (DCM), heterogeneous cardiomyopathy, is a significant cause of heart failure and heart transplant. Authors revealed the potential immune mechanism, 20 biomarkers, and therapeutic targets of DCM never published before.
Several concerns
- The authors identified DEGs first and applied it to Cibersort. In this case, it is thought that a deletion of the candidate gene group will occur because the immune cell portion is relatively small. Applying Cibersort using the entire dataset seems to fit the concept of this research.
- There is a question as to whether the finally selected genes can be applied only to DCM. Diseases that cause inflammation in the heart include not only DCM, but also ICM and ARVC. In ICM or ARVC, what changes do the authors expect to see in the genes described in this paper?
Author Response
Dear Reviewer
We are very grateful to Reviewer for reviewing the paper so carefully. These comments helped us improve the quality of the manuscript. We have addressed all comments and revised the manuscript accordingly.
Point 1:The authors identified DEGs first and applied it to Cibersort. In this case, it is thought that a deletion of the candidate gene group will occur because the immune cell portion is relatively small. Applying Cibersort using the entire dataset seems to fit the concept of this research.
Response 1: Thank you for your valuable suggestion. We agree with you that a deletion of the candidate gene group could occur in our study. However, using the entire dataset to perform CIBERSORT in our study may mask the role of some important genes. The genes with particularly low expression will influence the result, which should be deleted from all gene expression profile. DEGs can reflect the changes in disease well. Therefore, we finally used DEGs in our study.
Point 2: There is a question as to whether the finally selected genes can be applied only to DCM. Diseases that cause inflammation in the heart include not only DCM, but also ICM and ARVC. In ICM or ARVC, what changes do the authors expect to see in the genes described in this paper?
Response 2: Thank you for pointing out this problem in manuscript. We have revised the discussion according to your comments (Line 270-281). Enriched pathways are different between ICM and DCM (Sweet et al., 2018). Cytoskeletal and immune pathways belong to ICM, while the adhesion pathway is enriched in DCM. ICM was excluded by angiography in GSE120895 and GSE17800 datasets, so the hub genes in ICM may not be upregulated. ARVC, as another disease causing inflammation in the heart, has more active inflammatory signaling compared to DCM (Chen L et al., 2017), but the the transcriptomic level is different from DCM (Gaertner et al., 2012). We are not sure the changes of hub genes in ARVC. Further experiments can clarify it.
Sincerely,

Reviewer 3 Report
Dear Authors,
In this manuscript entitled “Identification of immune markers in dilated cardiomyopathies with heart failure by integrated weighted gene co-expression network analysis” Authors identified, using gene expression data from the GEO, immune-related biomarkers of dilated cardiomyopathies with heart failure. A group of six hub genes has been chosen and using an independent data set their diagnostic value was verified. Selected genes had good diagnostic power and can be used as potential diagnostic biomarkers and therapeutic targets for dilated cardiomyopathies. The paper addresses an important topic which is dilated cardiomyopathies, and identification of gene expression profile involved in immune mechanisms. However, during the review of this manuscript though, some remarks and comments appeared.
- The text of the manuscript should be corrected by a native English
- According to https://www.ncbi.nlm.nih.gov/geo/ GEO is the abbreviation for Gene Expression Omnibus.
- More detailed characteristic of the analyzed microarray data could give valuable information.
- All abbreviations used in the Figure 1 should be explained in the figure legend.
- Line 118: Volcano plot should be used in the manuscript instead of volcano.
- All titles of Figures are misleading and do not exactly reflect what the Figures show. The figure legends should be more detailed.
- All Figures have to be revised to have better font resolution.
- Human gene symbols should be italicized.
- Section 3.5. Full hub genes names should be given in the text.
- If possible, the direction of gene expression changes for hub genes and statistical significance (p-value) should be given in the text.
- Data obtained from in silico analysis for individual hub genes should be confirmed experimentally using clinical samples.
- Statistical significance (p-value) should be provided in the ROC curves analysis of hub genes, and marked in the Figure 8.
Yours faithfully,
Author Response
Point 1: The text of the manuscript should be corrected by a native English
Response 1: Thanks for your suggestion. We have polished the language in the revised manuscript.
Point 2:According to https://www.ncbi.nlm.nih.gov/geo/ GEO is the abbreviation for Gene Expression Omnibus.
Response 2: Thanks for your suggestion. We are very sorry for the mistake we made. And we have changed to “Gene Expression Omnibus” in the revised manuscript (Line 17, 65).
Point 3: More detailed characteristic of the analyzed microarray data could give valuable information.
Response 3: Thank you for your valuable suggestion. We gave more detailed characteristic of the analyzed microarray data in the revised manuscript (Line 73-79). Both GSE120895 and GSE17800 comprised DCM patients with left ventricular systolic dysfunction (LVEF<45%) and symptoms of HF according to the New York Heart Association (NYHA) classifications â…¡ and â…¢.
Point 4:All abbreviations used in the Figure 1 should be explained in the figure legend.
Response 4: We gratefully appreciate for your valuable comment. We are very sorry for our negligence. We have added the explanation for all abbreviations in the figure legend (Line 116-119).
Point 5: Line 118: Volcano plot should be used in the manuscript instead of volcano.
Response 5: Thank you for your valuable suggestion. We are very sorry for our incorrect writing and “volcano plot” was used in the revised manuscript instead of “volcano”(Line 127).
Point 6: All titles of Figures are misleading and do not exactly reflect what the Figures show. The figure legends should be more detailed.
Response 6: Thank you for your valuable suggestion. We are very sorry for our negligence. We changed all titles of Figures and gave more detailed information in the revised manuscript. Here are the new titles : Figure 1: Study flowchart (Line 115); Figure 2. Volcano plot and heatmap for differentially expressed genes identified in the GSE120895 dataset (Line 130); Figure 3. Gene Set Enrichment Analysis (GSEA) and Kyoto Encyclopedia of Genes and Genomes (KEGG) pathway enrichment analysis for GSE120895 (Line 141); Figure 4. Immune infiltration analysis for GSE120895 (Line 152); Figure 5. Weighted gene co-expression network analysis (WGCNA) for GSE120895 (Line 168); Figure 6. The scatter plots of module membership and gene significance for B cells and NK cells in the key modules (Line 176); Figure 7. Key modules that cotain hub genes of the PPI network (Line 181); Figure 8. Receiver Operating Characteristic (ROC) curves of hub genes in GSE17800 (Line 197); Figure 9. GSE17800 was used to verify the expression levels of hub genes in DCM and control (Line 201).
Point 7: All Figures have to be revised to have better font resolution.
Response 7: Thanks for your suggestion. We are very sorry for substandard figures. We have made new figures to get better font resolution to get in the revised manuscript (Figure 1, Line 116; Figure 2, Line130; Figure 3, Line 141; Figure 4, Line 152; Figure 5. between line 167 and 168; Figure 6, Line 176; Figure 7, Line 181; Figure 8, Line 197; Figure 9, Line 201). We deleted Figure 4C, Figure 5F and Figure 8A because of duplicate content or poor significance. We made a Figure 9 to verify the expression levels of hub genes in DCM and control.
Point 8: Human gene symbols should be italicized.
Response 8: Thank you for your valuable suggestion. We are very sorry for our negligence. All human gene symbols were italicized in the revised manuscript.
Point 9 : Section 3.5. Full hub genes names should be given in the text.
Response 9: Thank you so much for your careful check. We are very sorry for our negligence. Full hub genes names of genes we discussed were given in the revised manuscript (Melanocyte proliferating gene 1 for MYG1, Line 251; Flotillin1 for FOLT1, Line 259; autophagy-related protein 13 for ATG13, Line 268).
Point 10: If possible, the direction of gene expression changes for hub genes and statistical significance (p-value) should be given in the text.
Response 10: Thank you for your valuable suggestion. GSE17800 was used to verify the expression levels of hub genes in DCM and control which is shown in Figure 9 (Line 201). All hub genes were upregulated (Line 196), out of which MYG1, FLOT1 and ATG13 (P <0.05) had significant diagnostic values.
Point 11: Data obtained from in silico analysis for individual hub genes should be confirmed experimentally using clinical samples.
Response 11: Thank you for your valuable suggestion. We are very sorry for that we have not confirmed hub genes experimentally by clinical samples. When we got the comments from you, we started collecting clinical samples. We planed to include DCM patients with systolic dysfunction and symptoms of HF according to NYHA functional class â…¡ and â…¢, but we only included one patient in 10 days, which does not meet the requirements for the number of experimental samples. We are very sorry for that we can’t finish the experiment within 10 days.
Point 12: Statistical significance (p-value) should be provided in the ROC curves analysis of hub genes, and marked in the Figure 8.
Response 12: Thank you for your excellent suggestion. We are sorry that we ignore the statistical significance (p-value) in the ROC curves analysis of hub genes at the first time. We calculated the P value of in the ROC curves analysis of all hub genes, which were shown in Figure 8 (Line 197). And we found that not only GPX1 (P>0.05), but also LINC00520 (P>0.05) and ZNF548 (P>0.05) didn’t have statistical significance, which can be also shown in Figure 9 (Line 201). Therefore, we deleted the discussion of LINC00520 and ZNF548. MYG1, FLOT1 and ATG13 (P <0.05), with good diagnostic values, may be potential diagnostic biomarkers and therapeutic targets.

Round 2
Reviewer 1 Report
The authors have addressed our comments but some figures are still not legible. The "sample" axis of Fig 4A or 2B, for example, needs rework.
Reviewer 2 Report
The authors have addressed all the issues raised.